# Boosting Second Harmonic Generation Efficiency and Nonlinear Susceptibility via Metasurfaces Featuring Split-Ring Resonators and Bowtie Nanoantennas

**DOI:** 10.3390/nano14080664

**Published:** 2024-04-11

**Authors:** Yuan-Fong Chou Chau

**Affiliations:** Centre for Advanced Material and Energy Sciences, Universiti Brunei Darussalam, Tungku Link, Gadong, Bandar Seri Begawan BE1410, Brunei; chou.fong@ubd.edu.bn

**Keywords:** metasurface, SHG, conversion efficiency, bowtie-shaped Ag nanoantenna, resonator, nonlinear optical phenomena

## Abstract

This work investigates a metasurface design to achieve remarkable second harmonic generation (SHG) conversion efficiency and enhance effective nonlinear susceptibility using the finite element method. The elements of the designed structure are composed of a rectangular split-ring resonator Ag film, a bowtie-shaped Ag nanoantenna, and a pair of Bi bars that induce nonlinear optical phenomena due to the nonuniform distribution of the electric and magnetic fields within the device surface. The simulation results agree perfectly with the theory and demonstrate outstanding achievements in terms of SHG conversion efficiency (*η*) and effective nonlinear susceptibility ($\left|\chi_{eff}^{\left(2\right)}\right|$). Specifically, the metasurface reaches a peak *η* value of $4.544\times 10^{-8}$ and an effective nonlinear susceptibility of $3.4\times 10^{4}$ pm/V. This work presents a novel and versatile design to achieve high *η* and $\left|\chi_{eff}^{\left(2\right)}\right|\;$in an SHG metasurface.

## 1. Introduction

Optical metasurfaces, consisting of two-dimensional nanoscale resonator lattices, embody the core concept of flat optics [1,2,3,4,5]. They present distinct possibilities for functional flat optics by facilitating control over light waves’ transmission, absorption, and reflection [6,7]. Metasurfaces enable significant miniaturization of structural dimensions and unlock novel functionalities previously unattainable [8]. Numerous phenomena have been successfully demonstrated in linear optics [9]. However, there is a growing demand for customized nonlinearities, and nonlinear metasurfaces are expected to meet this requirement [10,11]. They are poised to enable nonlinear geometric Berry phase phenomena [12], nonlinear optical chirality [13], and nonlinear wavefront engineering [14,15].

Nonlinear optics has flourished since the initial observation of the second harmonic generation (SHG) in 1961 [16]. It is crucial in various optical devices [17,18,19,20]. Frequency conversion processes, such as harmonic generation, are fundamental in nonlinear optics because natural materials inherently exhibit weak nonlinear responses [15,21]. This fundamental principle guides exploring interactions between light and matter in the nanoscale dimension. It offers insight into the nonlinear optical processes in metal nanoparticles or dielectric nanostructures. These resonances result in various modes, including surface plasmon modes, multiphoton absorption modes, nonlinear wave mixing modes, toroidal modes, higher-order anapole modes, magnetic quadrupoles, nonlinear optical Kerr effects modes, nonlinear optical switching modes, and electric dipoles [22,23,24,25]. In general, SHG applications span a wide range of disciplines, highlighting its versatility and importance in modern optics and photonics research [26].

SHG is a nonlinear optical phenomenon with applications that span various research domains [27,28,29]. One major constraint in integrating noble metals and dielectric materials in plasmonic designs for nonlinear metasurfaces is the efficiency of nonlinear conversion and the effective susceptibility [30]. Various design approaches have been developed to enhance the nonlinear performance of plasmonic and hybrid metasurfaces [31], primarily relying on the resonant coupling of metaatoms [32]. However, the overall SHG conversion efficiency (*η*) in these metasurfaces has traditionally been relatively low in simulation works, typically in the 10^−12^∼10^−10^ [31,33,34]. Although the overall *η* value can reach 10^−5^ in experiments (e.g., ref. [35]), there is a significant difference in the overall SHG conversion efficiency results between theory and experiment due to the different experimental conditions and computational methods. Furthermore, the effective nonlinear susceptibility ($\left|\chi_{eff}^{\left(2\right)}\right|$) in the reported articles is often relatively low. Various optical modes have been explored in pursuit of a higher value of *η* in nonlinear harmonic generation and increased the $\left|\chi_{eff}^{\left(2\right)}\right|$. These modes include optically induced magnetic dipole resonances, higher-order multipoles, and composite resonances [36].

There has been a recent resurgence in scientific interest in the quadratic nonlinearities of metallic nanostructures. This resurgence can be attributed to the significant localization of electromagnetic fields induced by plasmonic oscillations of conduction electrons within plasmonic materials [37]. In particular, SHGs have been experimentally observed in various geometric configurations, including sharp metal tips [38], fishnet structures [39], split-ring resonators [40], multilayer photonic band gaps [41], and T-shaped [34] and L-shaped nanoparticles [42]. Suresh et al. have proposed alternative methods to improve nonlinearity, utilizing epsilon-near-zero materials with close to zero epsilon temperatures [43]. Among these structures, split-ring resonators (SRRs) have garnered particular attention because of the presence of fundamental plasmonic resonance, resulting in the solid localization of both electric and magnetic fields within the resonator gap of SRRs. This phenomenon, in turn, significantly improves the efficiency of the SHG. Consequently, the overall shape of the plasmonic materials established on the metasurface significantly determines the SHG response. For example, Klein et al. experimentally measured SHG from the metasurface formed by SRRs and found that the conversion efficiency can reach 10^−11^ [34]. Zeng et al. studied U-shaped SRRs and obtained an *η* of 6.6 × 10^−11^ [44]. Considering the remarkable progress in this emerging field of nonlinear optics, it is reasonable to anticipate the emergence of a new category of nonlinear photonic metadevices in the coming years. These devices could find applications in high-speed switching, entangled photons, supercontinuum generation, and nonlinear imaging [45,46].

This work presents a versatile design to obtain high total *η* and elevated $\left|\chi_{eff}^{\left(2\right)}\right|$ in an SHG metasurface using the finite element method (FEM). The total conversion efficiency (*η*) refers to the overall effectiveness of a metasurface in converting incident electromagnetic energy from one frequency to another. It encompasses all processes involved in frequency conversion, including absorption, reflection, and transmission, as well as nonlinear optical effects such as SHG. It quantifies the proportion of incident energy successfully converted to the desired output frequency, considering losses and nonlinear interactions within the metasurface structure. The proposed metasurface comprises an Ag rectangular film, a bowtie-shaped Ag nanoantenna, and a pair of Bismuth (Bi) bars that form a coupled system, thereby generating nonlinear optical effects. The distinctive characteristics of Bi conductors, such as their high nonlinear susceptibility, broad response range, compatibility with metasurface geometry, and low losses, collectively contribute to the observed enhancements in SHG efficiency [47,48]. Harnessing these properties facilitates the development of metasurfaces capable of efficient frequency conversion, thereby improving performance across a broad spectrum of wavelengths [49].

Consequently, they emerge as promising contenders for various nonlinear optical applications [50,51]. The metamaterials studied consist of planar arrays of SRRs. Each SRR can be likened to a miniature LC oscillator circuit. In this configuration, the open ends of a rectangular silver (Ag) ring constitute the capacitance denoted as *C*, while the ring itself represents a fraction of one winding of a magnetic coil, contributing to the inductance *L*. The magnetic dipole moment that emerges is aligned perpendicular to the plane of the metasurface. The interaction of the coupled localized electric and magnetic fields within the device causes the free electrons to oscillate in an anharmonic manner under the influence of the magnetic force, resulting in nonlinear effects. According to the theory, a high level of conductivity in the bi-bar pair, the Ag rectangular ring, and the Ag bowtie nanoantenna would be conducive to the SHG procedure. This is why the proposed metasurface can achieve outstanding *η* and highly effective nonlinear $\left|\chi_{eff}^{\left(2\right)}\right|$ simultaneously.

In contrast to previous studies, this research explores a promising strategy in nonlinear optics employing metasurfaces incorporating split-ring resonators and bowtie nanoantennas to achieve high *η* and $\left|\chi_{eff}^{\left(2\right)}\right|$. Moreover, the proposed metasurface allows for a straightforward implementation in experimentation and fabrication. It is essential to note that the device’s size can significantly influence its nonlinear optical properties, showing great promise for a diverse range of applications and holding significant potential for advancing nonlinear optics. The proposed design could be primarily linked to the microwave frequency range. Its relevance to the field of nanomaterials can be understood through the advanced fabrication techniques involved, the nanoscale features of the structures, the properties of the materials used, and the potential for cross-disciplinary applications in nanotechnology-related fields.

## 2. Materials and Methods

Figure 1 illustrates the schematic diagram of the unit cell within the proposed metamaterial, providing an overview of the design of the electromagnetic coupling metasurface. This metasurface comprises a pair of Bi bars and an Ag bowtie enclosed by an outer rectangular Ag ring, which serves as an SRR with cut-wire patterns deposited on an FR-4 substrate. We chose Ag as the plasmonic material because of its low metal loss and cost-effectiveness. It is worth mentioning that other noble metals (e.g., Au, Al, or Cu) can replace the plasmonic material, and the choice of the dielectric material, Bi, is solely driven by its exceptional conductivity and mobility, which significantly amplifies the metasurface’s nonlinear response. Furthermore, FR-4 was chosen due to its compatibility with the chip substrate.

Several structural dimensions are of significance in this context: *L*_1_ represents the length of the rectangular Ag ring, *L*_2_ is the length of the Bi bar, w_1_ is the width of the rectangular Ag ring, w_2_ is the width of the Bi bar, *g* is the gap between the rectangular Ag cut-wire segment, *d* is the gap between the rectangular Ag ring and the Bi bar, and the gap between the Bi bar and the Ag bowtie, and *P* is the period of the metasurface array. Furthermore, *t*_Ag_, *t_Bi_,* and *t*_FR-4_ are Ag, Bi, and FR-4 thicknesses. Furthermore, it should be noted that the bottom length and height of the Ag bowtie are determined as follows: the bottom length is indicated as “*L*_3_”, and the height is indicated as *h* = “*L*_1_/2 − w_1_ − w_2_ − 2*d*”. These parameters play a crucial role in defining the characteristics of the proposed metasurface. In the figure, key elements are labeled, including the local magnetic field ($\overset{⃑}{B}\left(\omega \right)$), electric field ($\overset{⃑}{E}\left(\omega \right)$), and magnetic force ($\overset{⃑}{F}\left(2\omega \right)$). The structure parameters have been fine-tuned by optimization using FEM, as shown in Table 1. The selection of a 100 nm thickness for Bi, thinner than the Ag thickness, was deliberately made to optimize the SHG performance of the device. By incorporating a thinner layer of Bi, we aim to enhance the nonlinear optical effects necessary for SHG while maintaining the structural integrity provided by the Ag layer. This design choice allows greater control over the SHG process and ultimately contributes to the overall efficiency and effectiveness of the device.

When an *x*-polarized electromagnetic (EM) wave impinges on the structure at normal incidence, it generates a circulating surface current. This current occurs at the resonance frequency of the proposed structure and subsequently generates a magnetic field perpendicular to the metasurface plane (see Figure 1a). As a result of this phenomenon, there are localized electric and magnetic fields within the device. These fields experience significant electromagnetic enhancement, often reaching hundreds of times the intensity of the incident field [52]. Concurrently, the electric field drives the motion of free electrons within the outer rectangular Ag ring, inner Bi bars, and the Ag bowtie on the metasurface. These electrons exhibit motion along the *x*-axis with a drift velocity represented as $\overset{⃑}{v}$. As the outer rectangular Ag ring and the inner Ag bowtie are positioned within the region of the enhanced magnetic field, a solid magnetic force emerges, acting perpendicularly to $\overset{⃑}{v}$. This circumstance culminates in applying a cumulative force upon the free electrons within the metasurface.

Within this metasurface design, the localized magnetic field is significantly enhanced within the outer rectangular Ag ring and the inner Bi bars and Ag bowtie nanoantenna on the metasurface. This augmentation substantially raises the prominence of the magnetic force. Given that the induced magnetic field aligns along the *z*-axis while the electric field aligns along the *x*-axis, the magnetic force induces oscillations of the electrons along the *y*-axis within the intersecting Ag surfaces, as depicted in Figure 1a. Consequently, it can be expected that the second harmonic wave (SH) emanates in both the +*z* and −*z* directions with *x*-polarization, as illustrated in Figure 1a. Of greater significance, this distinctive nonlinearity originates fundamentally from the magnetic force rather than the composite materials’ properties. This fundamental distinction distinguishes the proposed structure from most nonlinear metamaterials previously reported nonlinear metamaterials [53,54,55].

Polarizations of first- and second-order polarizations (${\overset{⃑}{P}}_{L}\left(\omega \right)$ and ${\overset{⃑}{P}}_{NL}\left(2\omega \right)$) can be expressed as follows [54]:(1)$${\overset{⃑}{P}}_{L}\left(\omega \right)=-\varepsilon_{0}\omega_{P}^{2}\overset{⃑}{G}\left(\omega \right)\overset{⃑}{E}\left(\omega \right),$$
(2)$${\overset{⃑}{P}}_{NL}\left(2\omega \right)=-\frac{\varepsilon_{0}\omega_{p}^{2}\mu_{e0}}{m^{*}}\;\overset{⃑}{G}\left(2\omega \right)\left|\overset{⃑}{B}\left(\omega \right)\right|\left|\overset{⃑}{E}\left(\omega \right)\right|{\hat{a}}_{y},$$

Here, $\varepsilon_{0}$ is the permittivity in air, $\omega_{p}$ is the plasma frequency$,{and\;\mu }_{e0}$ is the direct current mobility of the material.

${\overset{⃑}{E}}_{0}\left(\omega \right)$ denotes the input electric field, while σ represents the Ag’s conductivity.$\overset{⃑}{\;B}\left(\omega \right)$ can be induced by ${\overset{⃑}{E}}_{0}\left(\omega \right)$. Following Ampere’s circuital law, i.e., $\overset{⃑}{B}\left(\omega \right)\propto \overset{⃑}{J}\left(\omega \right)=\sigma {\overset{⃑}{E}}_{0}\left(\omega \right)$.

Moreover, the increased conductivity inherent to the material composing the SRR in the outer rectangular Ag ring and the inner Ag bowtie nanoantenna yields advantages for the intensity of SHG. This enhanced conductivity improves the current density, thereby strengthening the magnetic field. As indicated in Equation (2), increasing the mobility of the metasurfaces within the rectangular outer Ag ring and the inner Ag bowtie would similarly increase the proportional SHG. This phenomenon results from electrons with enhanced mobility achieving higher drift velocities when subjected to the same electric field, leading to a more significant magnetic force.

## 3. Results

### 3.1. Frequency Domain Results

Frequency domain results are required before the study of nonlinear effects using time domain simulations. They help to understand the optical response, identify relevant resonant modes, provide initial conditions for nonlinear simulations, improve computational efficiency, and validate the simulation setup. Therefore, obtaining results before studying nonlinear effects is crucial in the simulation workflow. Simulations are carried out using the COMSOL Multiphysics software (version 6.0), employing FEM. First, a calculation of the transmittance (T(ω)), reflectance (R(ω)), and absorptance (A(ω) = 1 − R(ω) − T(ω)) spectra of the device is carried out based on the frequency domain. In the frequency domain, the S parameters of the proposed device were subjected to FEM simulations in the range of 5 to 15 GHz, with increments of 0.1 GHz, where R(ω) = |S_11_|^2^ and T(ω) = |S_21_|^2^ signify reflectance and transmittance.

The structural parameters shown in Figure 1b were fine-tuned for optimal performance within the microwave range to produce a significant intensity of SHG. A simulation of a single unit cell was conducted to emulate the periodic array structure. The σ of Ag and Bi are defined as 6.30 × 10^7^ S/m and 2.2 × 10^7^ S/m, while the permittivity of FR4 is 4.2 + *i* 0.1 using the frequency domain within the calculated frequency, respectively. In addition, the mobility of a pair of Bi bars is 0.11 m^2^/V·s [54]. Ag and Bi are material settings that use transient boundary conditions in COMSOL multiphysics simulations.

The spectrum depicted in Figure 2a,b initially simulates the reflectance, transmittance, and absorptance spectrum of two cases: one without the Ag bowtie surface (referred to as Case 1) and the other with the Ag bowtie surface (referred to as Case 2). These spectra are crucial for understanding the system’s behavior under study, particularly in the context of nonlinear wave processes. Observing the transmittance and reflectance spectrums in both cases reveals essential insights. Notably, a dip in transmittance and a peak in reflectance are observed around 10 GHz in both cases, coinciding with the system’s resonant frequency. This indicates strong interactions between the incident electromagnetic waves and the structure at this frequency. These resonant features suggest that the system exhibits significant nonlinear response capabilities, particularly near the resonant frequency. The localization of the magnetic field at around 10 GHz further emphasizes the importance of this frequency range for nonlinear wave processes. Moreover, the differences in transmittance and reflectance peaks and dips between Case 1 and Case 2 highlight the impact of the Ag bowtie surface on the system’s optical properties. These differences can inform the design and optimization of the structure for specific nonlinear applications, enabling tailored manipulation of nonlinear wave processes.

The reflectance and transmittance spectra provided in Figure 2a,b provide valuable information on the resonant behavior of the system and its potential for nonlinear wave processes. Understanding this spectrum is crucial for analyzing and optimizing the system’s nonlinear response, guiding the development of advanced photonic devices with enhanced functionality.

The surface current and magnetic field distributions depicted in Figure 3a,b provide crucial insights into the system’s behavior at 10 GHz for both Case 1 and Case 2. These visual representations are invaluable for analyzing nonlinear wave processes and understanding the mechanisms underlying enhanced nonlinear optical effects. In both cases, circulating currents are observed, creating heightened magnetic fields around specific structure regions. In Case 1, the currents mainly concentrate around the inner edge of the rectangular Ag ring, while in Case 2, they extend to both the inner edge of the rectangular Ag ring and the gap within the metasurfaces. These circulating currents exhibit a counterclockwise direction on the surfaces of the Ag ring, the Ag bowtie nanoantenna, and the Bi bars, reaching their maximum strength near the gaps between the outer rectangular ring and the Bi bars. A comparison between Case 1 and Case 2 reveals the significant impact of the Ag bowtie nanoantenna surface on the localized $\overset{⃑}{\;B}\left(\omega \right)\;$intensity. In Case 2, the Ag bowtie nanoantenna surface enhances the magnetic field intensity on both sides of the Bi bars, primarily attributed to the gap between the surface of the Ag bowtie nanoantenna and the Bi bar. The magnetic field intensity enhancement in Case 2 surpasses that of Case 1, indicating the effectiveness of the Ag bowtie nanoantenna surface in boosting nonlinear optical effects. These findings are crucial for understanding and optimizing the system’s nonlinear response. The enhanced magnetic field intensity in Case 2 (reaching a maximum strength of 62.40 times stronger than the incident field at 10 GHz), facilitated by the Ag bowtie nanoantenna surface, suggests improved nonlinear optical phenomena such as SHG. This enhancement underscores the importance of the Ag bowtie nanoantenna surface in tailoring and optimizing the system for advanced nonlinear optical applications.

### 3.2. Time Domain Results

We conducted time domain simulations to investigate the electric and magnetic coupling interactions in Case 1 and Case 2 and used the same structural parameters in Table 1. In the time domain, the incident electric field is described as *E* = (*E*_x_, 0, 0), with *E*_x_ expressed as follows:(3)$$\overset{⃑}{E_{x}}=\overset{⃑}{E}\left(\omega \right)\cos\left(\omega t-k_{0}z\right)e^{{-(\frac{t-t_{0}}{∆t})}^{2}},$$

In this Equation, $\overset{⃑}{E}\left(\omega \right)\;isthepeakamplitudeofthe\overset{⃑}{E_{x}}$, *ω* is the angular frequency, *t* is time, *k*_0_ is the wave number, *z* is the coordinate of the *z* axis, and Δ*t* represents the parameters governing the Gaussian pulse. In simulations, we used $\overset{⃑}{E}\left(\omega \right)=10^{7}\;\;V/m$, $\omega ={2\pi \times 10}^{10}\frac{rad}{s}$, *t*_0_ = 1.6 ns, Δ*t* = 600 ps and the simulation covered a total period of 4 ns, with a time step of 1 ps [54].

Ag and *Bi* material properties are represented as surface current densities on the surface of the proposed device [56]. The *σ*(*ω*) of the *Bi* bars based on the Drude model can be described as Equation (4) [54]:(4)$$\sigma \left(\omega \right)\approx \sigma_{0}\left[\begin{array}{ccc} \frac{1}{1+{\left(\mu_{Bi}B\right)}^{2}} & \frac{-\mu_{Bi}B}{1+{\left(\mu_{Bi}B\right)}^{2}} & 0\\ \frac{\mu_{Bi}B}{1+{\left(\mu_{Bi}B\right)}^{2}} & \frac{1}{1+{\left(\mu_{Bi}B\right)}^{2}} & 0\\ 0 & 0 & 1 \end{array}\right],$$

The ratio of the amplitudes of the $\left|\overset{⃑}{F_{B}}\right|$ to the $\left|\overset{⃑}{F_{E}}\right|$ (denoted as $R_{m/e})\;$can express as follows:(5)$$R_{m/e}=\frac{\left|\overset{⃑}{F_{B}}\right|}{\left|\overset{⃑}{F_{E}}\right|}=\frac{q\left|\overset{⃑}{v}\right|\left|\overset{⃑}{B}(\omega )\right|}{q\left|\overset{⃑}{E}(\omega )\right|}=\frac{\left|\overset{⃑}{v}\right|\left|\overset{⃑}{B}(\omega )\right|}{\left|\overset{⃑}{E}(\omega )\right|}=\frac{\mu_{e0}\left|\overset{⃑}{E}(\omega )\right|\left|\overset{⃑}{B}(\omega )\right|}{\left|\overset{⃑}{E}(\omega )\right|}=\mu_{e0}\left|\overset{⃑}{B}(\omega )\right|=\frac{{N\;\mu }_{e0}\left|\overset{⃑}{E_{0}}\left(\omega \right)\right|}{c},$$
where $\left|\overset{⃑}{v}\right|$ is the drift velocity, *c* is the speed of light, and *N* is the enhanced time of the magnetic field. Based on Equation (5), the high conductivity of the *Bi*-bar pair, the Ag rectangular ring, and the Ag bowtie nanoantenna would favor the SHG efficiency.

Substituting the values of $c=3\times 10^{8}\;m/s$, *N* = 62.40 (the enhanced time of the magnetic field based on Figure 3b), $E_{0}=10^{7}\;V/m$, and${\;\mu }_{e0}=0.11\;m^{2}/V\cdot s$ (the electron drift mobility on *Bi*) [57], we obtain the $\left|\overset{⃑}{F_{B}}\right|\;$= 62.40 × 0.11 × 10^7^/3 × 10^8^ = 22.88% of the $\left|\overset{⃑}{F_{E}}\right|$, a significant enough difference to induce a nonlinear response [48,58] and this result perfectly matches the theory.

Figure 4 and Figure 5 compare the transmitted and reflected electric fields at 10 GHz for Case 1 (Figure 4a–d) and Case 2 (Figure 5a–d), respectively. Compared to the incident fundamental wave, the double frequency in Figure 4 and Figure 5 demonstrates that the proposed metasurface successfully generates the SH wave at 20 GHz.

In Case 1, the transmission spectrum (see Figure 4a) and reflection spectrum (see Figure 4c) in the time domain and the frequency spectrum in Figure 4b,d, obtained by the Fourier transformation, show an apparent SHG effect. In the Fourier transformation used in the research, the input refers to the time domain signal representing the electromagnetic field distribution, typically generated by numerical simulations. On the other hand, the output refers to the frequency domain spectrum obtained by transforming the input signal from the time domain to the frequency domain. This spectrum provides valuable information about the frequency components in the electromagnetic field, allowing for the analysis of resonance phenomena and nonlinear effects. The transmission and reflection powers for Case 1 at 20 GHz reach 46.57 W and 26.22 W, respectively. The calculated incident wave power is $1.944\times 10^{10}$ W, and the total conversion efficiency (*η*), defined as *η* = (power of transmission + power of reflection transmission)/input power of the metasurface at 20 GHz, is calculated as η= (46.57 W + 26.22 W)/$1.944\times 10^{10}$ W = 3.733 $\times \;10^{-9}$.

The effective nonlinear susceptibility ($\left|\chi_{eff}^{\left(2\right)}\right|$) can be calculated using Equation (6) [59]:(6)$$\left|\chi_{eff}^{\left(2\right)}\right|=\frac{c\sqrt{\eta }}{\omega t_{Ag}\left|\overset{⃑}{E}\left(\omega \right)\right|}$$
where *η* represents the conversion efficiency of SHG.

In Case 1, the nonlinear process arises from the coupling between the rectangular ring and the Bi bar metasurface. By substituting the parameters into Equation (6), which include$s\eta =3.733\times 10^{-9},\;\left|\overset{⃑}{E}\left(\omega \right)\right|=10^{7}\;V/m$, $\omega =2\pi \times 10^{10}$ rad/s, $c=3\times 10^{8}\;$m/s, and *t*_Ag_ = 30 μm, results in $\left|\chi_{eff}^{\left(2\right)}\right|=0.972\times 10^{3}$.

In Case 2, the transmission spectrum (see Figure 5a), the reflection spectrum (see Figure 5c), and the frequency spectrum in Figure 5b,d, obtained through the Fourier transformation, exhibit a significantly more pronounced SHG effect than in Case 1. The transmission and reflection powers for Case 2 at 20 GHz reach 504.50 W and 286.00 W, respectively. Consequently, the total conversion efficiency (*η*) of the metasurface at 20 GHz is η= (504.50 W + 286.00 W)/$1.944\times 10^{10}$ W = 4.07 $\times \;10^{-8}$. This value is one order of magnitude higher than that of Case 1 and significantly surpasses the levels reported in other literature, such as [33,34,35].

Because of the fundamental plasmonic resonance’s emergence, electric and magnetic fields become highly concentrated within the gaps of the Ag rectangular ring, the Ag bowtie nanoantenna, and the Bi bars. Consequently, this significantly increases the efficiency of the SHG. Using Equation (6), we can obtain $\left|\chi_{eff}^{\left(2\right)}\right|=3.21\times 10^{4}$ pm/V. This value is 33.1 times higher than that of Case 1 and 26.8 times higher than that reported in ref. [54].

Discussing how variations in structural parameters affect resonance frequency would provide deeper insight into the tunability and optimization of the proposed metasurface design [17,60]. The resonance frequency of a metasurface is intricately linked to its structural parameters. Variations in parameters such as *L*_3_, *g*, *d*, and w_2_ directly influence the device’s optical properties and electromagnetic response [61,62]. For example, changes in *L*_3_ and *g* can alter the effective refractive index and the electromagnetic coupling between adjacent resonators, affecting the resonant behavior of the metasurface. Similarly, adjustments in *d* can modify the propagation constants of the guided modes supported by the substrate, influencing the resonance conditions of the metasurface.

Furthermore, variations in w_2_ can affect the effective mode volume and modal distribution within the nonlinear material, consequently affecting the resonance frequency and the efficiency of nonlinear optical processes such as SHG. This work can provide valuable information on the design principles and optimization strategies for tailoring metasurfaces to specific applications by analyzing how variations in these structural parameters affect the resonance frequency. Understanding the tunability of the resonance frequency enables precise control over the optical response of the metasurface, facilitating the development of advanced photonic devices with enhanced performance and functionality.

The length (*L*_3_) of the Ag bowtie metasurface and the rectangular split gap of the Ag rectangular ring (*g*) play a crucial role in influencing the effect of SHG in the proposed structure. Figure 6a,b illustrate the variation in the relationship between the *L_3_* and *g* values and the total conversion efficiency, *η*. The structural parameters remain the same as in Table 1, with one parameter being varied, while the others remain constant. In Figure 6a, it can be observed that *η* is $3.733\times 10^{-9}$ when *L*_3_ = 0.0 mm (i.e., Case 1), which increases to a peak value of $\eta =4.544\times 10^{-8}$ when *L*_3_ = 0.9 mm and decreases to $\eta =3.566\times 10^{-8}$ when *L*_3_ = 1.2 mm. These results can be explained by the different coupling effects between the Bi bar surfaces and the Ag bowtie surface under varying lengths (*L*_3_) of the Ag bowtie film. The peak value of $\eta =4.544\times 10^{-8}$ when *L*_3_ = 0.9 mm corresponds to $\left|\chi_{eff}^{\left(2\right)}\right|=3.4\times 10^{4}$ pm/V. This value is 34.98 times greater than that of Case 1 and 28.33 times higher than that reported in ref. [54]. Furthermore, the $\left|\chi_{eff}^{\left(2\right)}\right|\;$obtained for the designed metasurface significantly exceeds that of the cases relying on metal nanostructures [40] and traditional nonlinear crystals [6,7].

In Figure 6b, it is evident that *η* = 0.0 when *g* = 0.0 mm (that is, there is no gap in the rectangular Ag ring), and this is because there is no SRR effect to support the enhancement of the electric field in the proposed metasurface [26,27]. As g increases from 0.0 to 0.0025 mm, the *η* value of abruptly increases from 0.0 to $3.455\times 10^{-8}$, indicating that the SRR effect begins to influence the proposed SHG system. The η value falls within the range of $3.836\times 10^{-8}<\eta <4.356\times 10^{-8}$ as *g* varies from 0.025 mm to 0.35 mm, reaching a maximum value of $\eta =4.356\times 10^{-8}$ when *g* = 0.3 mm. However, the value of *η* decreases from $4.205\times 10^{-8}$ to $3.317\times 10^{-8}$ when *g* changes from 0.4 mm to 1.1 mm. These results can be attributed to the different SRR effects induced within the rectangular split gap of the Ag rectangular ring.

The nonlinear response is intimately linked to the intensity of the electric and magnetic fields, which exhibit nonuniform features within the Case 2 metasurface. This nonuniformity is particularly evident in the gap region between the rectangular Ag ring, Ag bowtie nanoantenna, and the Bi bars, as Figure 7a shows. Using the structural parameters in Table 1, we have changed one parameter at a time while keeping the others fixed.

In this context, it is crucial to highlight that the optimal gap distance between the Ag surfaces and the Bi bars, where the electromagnetic field is stronger, significantly increases SHG intensity. Specifically, within the range of *d* = 0.04 to 0.08 mm, a notable trend is observed. As *d* varies from 0 to 0.06 mm, the *η* increases from 0 to its peak value of $4.18\times 10^{-8}$. Subsequently, as *d* continues to vary from *d* = 0.06 mm to *d* = 0.08 mm, the *η* decreases to$\;2.21\times 10^{-8}$. It is worth noting that the *η* value consistently remains above$\;4.05\times 10^{-8}$ when the *d* value falls within the range of 0.04–0.06 mm.

Figure 7b illustrates how the increase in the width of the Bi bar (w_2_) improves the strength of the SHG. As w_2_ varies from 0 mm to 0.8 mm within the range of $7.92\times 10^{-11}<\eta <1.34\times 10^{-8}$, a noteworthy trend emerges. Specifically, as w_2_ increases, the strength of SHG shows a significant rise, culminating at $4.11\times 10^{-8}$ when w_2_ = 0.5 mm. This observation suggests that a suitable w_2_ allows more free electrons to interact with the localized magnetic field, resulting in a more pronounced nonlinear response.

The observed sensitivity of the conversion efficiency to specific values of the width of the Bi bar (w_2_) can be attributed to several factors. Firstly, the width of the Bi bar directly influences the optical confinement within the device structure, affecting the intensity of incident light that interacts with the nonlinear material. The variation in w_2_ alters the optical mode profile and the overlap between the optical field and the Bi layer, consequently affecting the efficiency of the SHG process. Second, changes in w_2_ can modify the dispersion properties of the waveguide, leading to variations in the phase-matching conditions crucial for efficient SHG. Slight alterations in w_2_ may result in significant shifts in the effective refractive index, affecting the phase matching between the fundamental and second harmonic frequencies. Moreover, the width of the Bi bar can influence the modal distribution of the electromagnetic field within the device, affecting the nonlinear polarization generated and, subsequently, the efficiency of the SHG. It would be valuable to include data on slight alterations in w_2_ in the work to provide a comprehensive understanding of this sensitivity. By systematically varying w_2_ and measuring the corresponding conversion efficiencies, we can elucidate the tolerance range of w_2_ and its impact on the device’s efficiency. These experimental data provide insight into the optimal design parameters to maximize the performance of SHG while providing practical device fabrication and optimization guidelines.

This work presents SHG results assuming an infinite array structure under an infinite-plane wave excitation. Incorporating discussions or results regarding how a finite beam size might affect SHG performance would enhance the study’s relevance to real-world applications where beam sizes are finite. Finite beam sizes can introduce additional complexities to the SHG process compared to the idealized scenario of infinite-plane wave excitation [63]. Firstly, the spatial profile of the incident beam can affect the intensity distribution within the nonlinear material, influencing the efficiency of SHG. For instance, Gaussian beam profiles, commonly encountered in practical applications, exhibit a nonuniform intensity distribution across the beam diameter, potentially leading to variations in the SHG efficiency along the beam axis. The finite beam size can also impact the phase-matching conditions that are crucial for efficient SHG. Spatially varying phase fronts across the beam profile may introduce phase mismatches within the nonlinear material, affecting the constructive interference necessary for efficient frequency conversion.

Furthermore, the interaction between the finite beam and the device structure can result in spatially varying optical modes, affecting the modal distribution of the electromagnetic field within the nonlinear material and, consequently, the efficiency of SHG. By including discussions or experimental results on the impact of a finite beam size on SHG performance, the study can provide valuable information on practical considerations and limitations relevant to real-world applications. Understanding how finite beam sizes influence SHG efficiency facilitates designing and optimizing nonlinear optical devices tailored for specific experimental setups and applications.

The resonators used in the proposed design are deep sub-wavelengths, which can pose challenges in fabrication processes because of the high precision required. Increasing the size of parameters such as period (*P*) and other dimensions may help mitigate some of these fabrication requirements by making the structures easier to fabricate. However, it is vital to maintain the overall unit cell of the metasurface and subwavelength to preserve its unique optical properties and functionalities. Increasing the size of individual resonators or other features while keeping the unit cell sub-wavelength allows for scalability and flexibility in design while achieving the desired nonlinear optical effects. Therefore, a careful balance needs to be struck between making the resonators easier to fabricate by increasing their size and maintaining the sub-wavelength nature of the overall metasurface unit cell to preserve its optical performance. This balance ensures that the metasurface remains practical for fabrication while achieving the desired enhancement in SHG efficiency and nonlinear susceptibility.

The absence of experimental data in the paper is a notable limitation. To address this, it is crucial to consider the required experimental setup (i.e., how the proposed metasurface would be implemented and tested in a real-world scenario), fabrication steps (i.e., how the metasurface structures would be manufactured, involving detailed fabrication techniques such as lithography, deposition, etching, or other nanofabrication processes, such as material selection, substrate choice, and post-processing steps), and potential risks (i.e., including fabrication imperfections, material nonuniformities, substrate effects, or environmental factors) that could affect performance compared to the designed simulations.

When comparing earlier works on deeply subwavelength resonators, such as those by Ali-ci et al. [64] and Serebryannikov et al. [65], with the design proposed in the present manuscript, several aspects emerge: (1) dominant physics: In [64,65], the dominant physics may involve the interaction of electromagnetic waves with deep subwavelength resonators, resulting in phenomena such as surface plasmon resonance (SPR) or Mie resonances. Both the articles mentioned above and this paper share a similar purpose, focusing on understanding the electromagnetic behavior of deep subwavelength structures and their impact on overall device performance. (2) Extent of miniaturization: This aspect involves assessing the depth of sub-wavelength resonators in each case and how this miniaturization contributes to enhancing device functionality, such as nonlinear optical effects or antenna performance. The proposed design in our manuscript enables significant miniaturization compared to the designs discussed in the papers by [64,65].

## 4. Conclusions

In summary, this study presents the development of a high-efficiency SHG metasurface with impressive conversion capabilities and high nonlinear susceptibility. The metasurface comprises an Ag rectangular and bowtie-shaped Ag nanoantenna coupled with a pair of Bi bars, and its performance is evaluated using the FEM. This design harnesses nonlinear SHG effects, as the nonlinear response is directly related to the distribution of the electromagnetic field within the metasurface, which is notably nonuniform. The metasurface induces coupled localized electromagnetic fields, leading to oscillations of free electrons in a nonharmonic manner due to magnetic forces, resulting in a notable nonlinear response. The critical structural parameters *L*_3_, *g*, *d*, and w_2_ profoundly influence this nonlinearity. The results indicate that the maximum η reaches $4.544\times 10^{-8}$ when *L*_3_ is set to 0.9 mm, corresponding to $\left|\chi_{eff}^{\left(2\right)}\right|=3.4\times 10^{4}$ pm/V. This value for *η* surpasses that reported in previous studies. Furthermore, the $\left|\chi_{eff}^{\left(2\right)}\right|$ achieved with this novel metasurface significantly outperforms that of previously reported nonlinear metamaterials that depend on metal plasmonic nanostructures and nonlinear crystals. This investigation significantly advances the design and comprehension of nonlinear metasurfaces with strong *η* and $\left|\chi_{eff}^{\left(2\right)}\right|$. The potential applications span a wide range and offer exciting possibilities in nonlinear optics.

## Figures and Tables

**Figure 1 nanomaterials-14-00664-f001:**
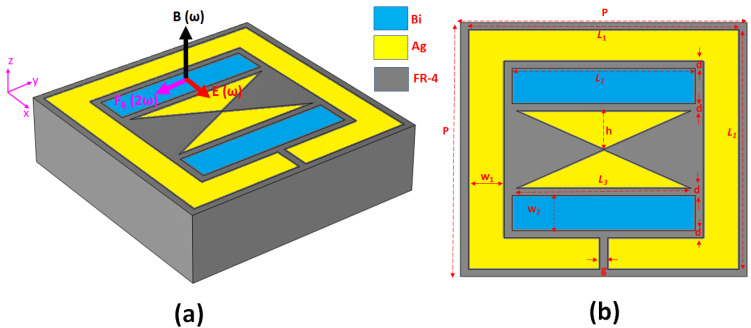
Schematic diagram of the unit cell and setting parameters: (**a**) 3D view, and (**b**) top view.

**Figure 2 nanomaterials-14-00664-f002:**
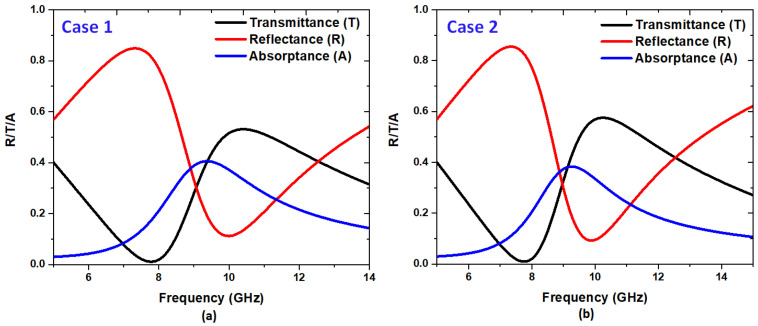
Transmittance, reflectance, and absorptance for (**a**) the case without the Ag bowtie surface (Case 1) and (**b**) the case with the Ag bowtie surface (Case 2).

**Figure 3 nanomaterials-14-00664-f003:**
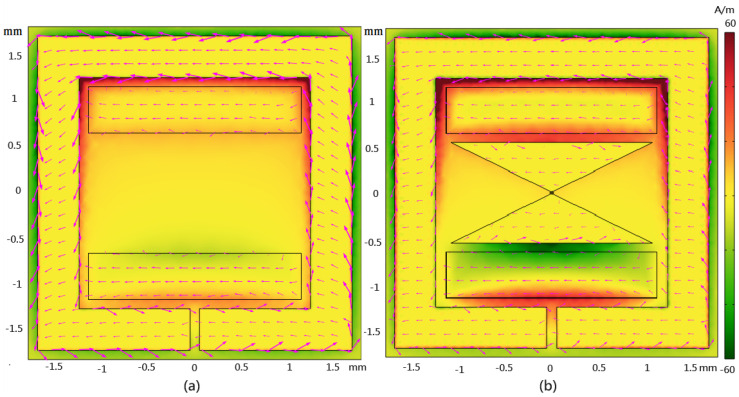
Magnetic field distributions for the unit cell, (**a**) magnetic field distribution for Case 1, and (**b**) magnetic field distribution for Case 2 at 10 GHz. Red arrows indicate surface currents.

**Figure 4 nanomaterials-14-00664-f004:**
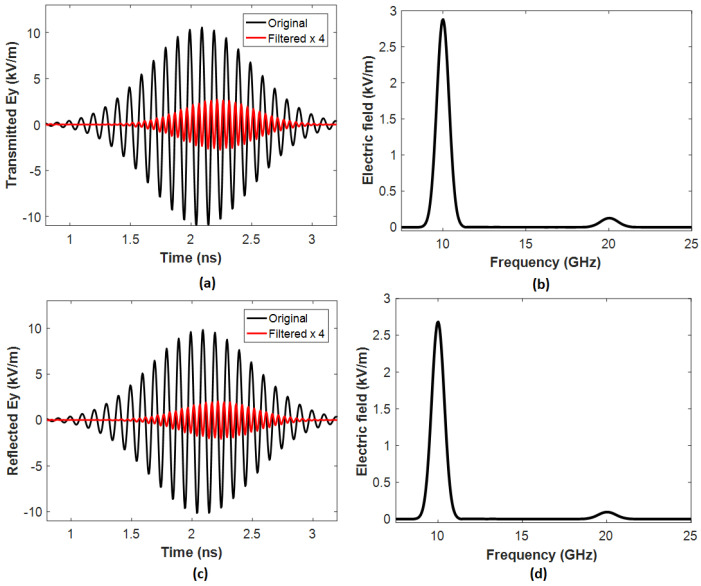
(**a**) Transmission and (**c**) reflection spectrum in the time domain of Case 1. The frequency spectrum of (**b**) transmission and (**d**) reflection obtained by the Fourier transformation.

**Figure 5 nanomaterials-14-00664-f005:**
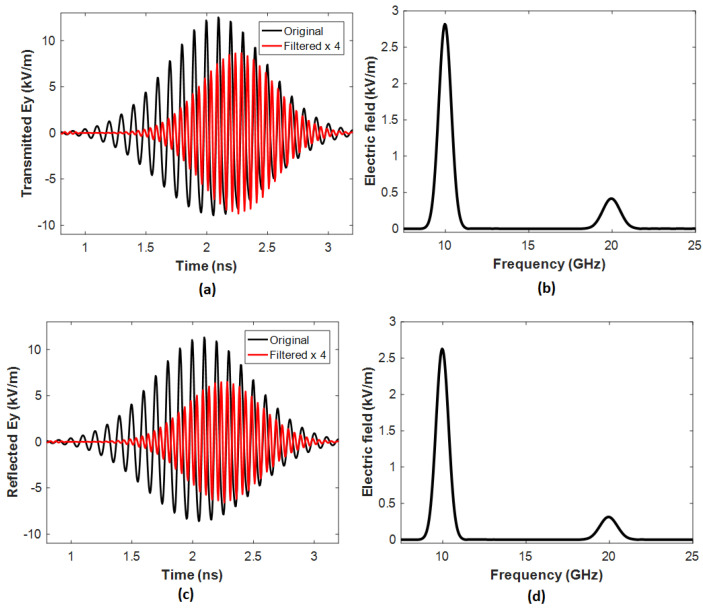
(**a**) Transmission and (**c**) reflection spectrum in the time domain of Case 2. The frequency spectrum of (**b**) transmission and (**d**) reflection obtained by the Fourier transformation.

**Figure 6 nanomaterials-14-00664-f006:**
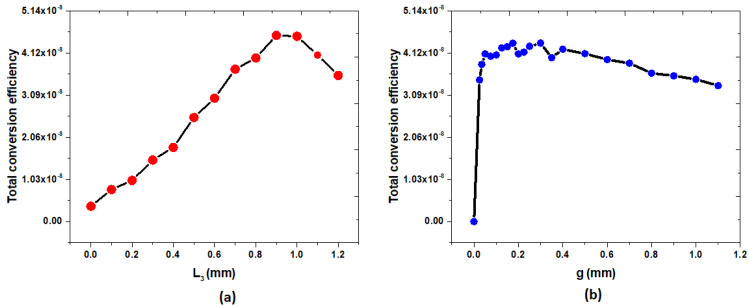
Relationship between (**a**) *L*_3_ and (**b**) *g* values and total conversion efficiency, η.

**Figure 7 nanomaterials-14-00664-f007:**
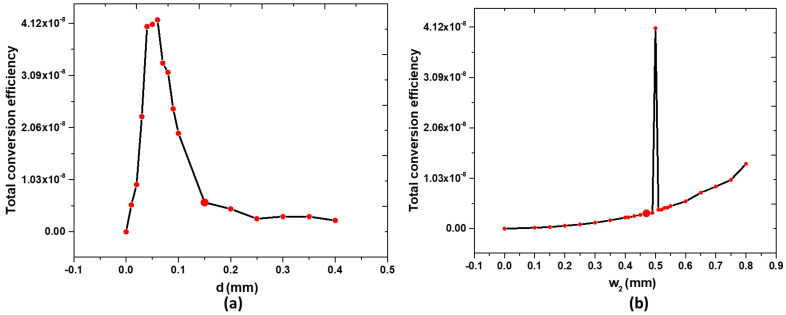
Relationship between (**a**) the values of *d* and (**b**) w_2_ and the total conversion efficiency, η.

**Table 1 nanomaterials-14-00664-t001:** The device’s parameters used in FEM simulations.

Name	Symbol	Value
the length of the rectangular Ag ring	*L* _1_	3.4 mm
the length of the Bi bar	*L* _2_	2.3 mm
the bottom length of the Ag bowtie	*L* _3_	2.2 mm
the height of the Ag bowtie	*h*	0.65 mm
thickness of Ag	*t* _Ag_	30 μm
thickness of Bi	*t* _Bi_	100 nm
thickness of FR-4	*t* _FR-4_	1 mm
the gap between the rectangular Ag ring and the Bi bar, and the gap between the Bi bar and Ag bowtie	*d*	0.05 mm
the width of the rectangular Ag ring	w_1_	0.45 mm
the width of the Bi bar	w_2_	0.5 mm
the gap between the rectangular Ag cut-wire segment	*g*	0.1 mm
the period of the metasurface array	*P*	3.6 mm

## Data Availability

Data are contained within the article.

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
