# Peer review of "Boosting Second Harmonic Generation Efficiency and Nonlinear Susceptibility via Metasurfaces Featuring Split-Ring Resonators and Bowtie Nanoantennas"

_nanomaterials, 2024, doi:10.3390/nano14080664_

Round 1

Reviewer 1 Report

Comments and Suggestions for Authors

This article proposes a design to significantly enhance the nonlinear susceptibility of a Metasurface and validates the design by very good modelling supported by excellent.

The introduction provides an excellent summary review of the state of the field with a good set of references that would be very helpful for students and early-carrier researchers.

I have found the “methodology” of “presentation” by the author to be excellent.

The author did not present any physical experiments. The author used a good popular computer modeling software (COMSOL Multiphysics), as many researchers have been doing for a couple of decades to check the theoretical model for its potential consistency. All that proves is that the author’s assumptions in constructing his theory and the built-in assumptions in the computer-software are mutually consistent with each other – both use the same accepted theory of Electromagnetism.

     Regarding “further controls”, I would leave that to the author for his next research paper regarding what other parameters he would like to adjust to generate better efficiency. The current modeled plots look good to me.

·         Inconsistency: The author’s claim of “impressive conversion capability” of η=4,544x10<-8> contradicts the value obtained by Shi η=2.0x10<-5> [Ref.#27]. You should request the author for clarifications.

·         Experiments: The author did not present any physical experiments. He presented computer SIMULATED plots and curves, which are quite common in this field to get “some publications”.

Reviewer 2 Report

Comments and Suggestions for Authors

The manuscript titled "Boosting Second Harmonic Generation Efficiency and Nonlinear Susceptibility via Metasurfaces Featuring Split-Ring Resonators and Bowtie Antennas" presents an approach to enhancing second harmonic generation (SHG) conversion efficiency and effective nonlinear susceptibility through the introduction of bismuth (Bi) bars and bowtie structures within split-ring resonator (SRR) arrays. The work utilizes Ag as the primary material for the SRR and bowtie structures, employing Bi due to its unique anisotropic conductivity properties. This combination, as analyzed using the finite element method, yields significant improvements in SHG conversion efficiency and effective nonlinear susceptibility, surpassing previous works.

However, the claim of an increase in conversion efficiency by up to three orders of magnitude, as mentioned in the abstract, requires careful consideration. This remarkable improvement is attributed to the anisotropic conductivity of the used Bi conductors, suggesting a comparison that may not be entirely equitable with previous studies that did not utilize materials with similar conductivity properties. Therefore, it is recommended that the manuscript be published after addressing the following issues.

Comments

1. The manuscript should address the concern regarding the direct comparison of conversion efficiency improvements. It is suggested that the authors provide a more detailed analysis, taking into account the unique properties of the Bi conductors and how these contribute to the observed efficiency improvements. This could help readers understand the extent to which the efficiency gain is a result of the innovative design versus the intrinsic properties of the materials used.

2. The choice of a 100 nm thickness for Bi, significantly thinner than the Ag thickness, is curious. An explanation or justification for this design choice should be added to the manuscript to clarify its importance to the overall device performance.

3. A minor typographical error is identified at Line 114: "Magnetic field (E), electric field (B) ...." --> "Electric field (E), magnetic field (B) ...."

4. The manuscript would benefit from a discussion on how variations in the structural parameters (L3, g, d, w2) affect the resonance frequency. Such analysis could provide deeper insights into the tunability and optimization of the metasurface design.

5. In Figure 7(b), it's observed that the conversion efficiency is particularly sensitive to certain values of w2. Could you elucidate the reasons behind this distinct sensitivity? Moreover, including data on slight alterations in w2 would be invaluable. Such insights would allow us to assess the tolerance range of w2 and its impact on the device's efficiency.

6. Lastly, the manuscript currently presents SHG results assuming an infinite array structure under an infinite plane wave excitation. Including results or a discussion on how a finite beam size might affect SHG performance could significantly enhance the relevance of the study to real-world applications, where beam sizes are finite.

Reviewer 3 Report

Comments and Suggestions for Authors

The paper is dedicated to numerical demonstration of nonlinearity enhancement by using the specially designed metasurfaces. The author suggests an improved design of microwave metasurface comprising the deeply subwavelength resonators, which makes possible stronger nonlinearity than in the earlier suggested designs. The paper can be interesting for journal reader but significant revision is needed at this stage.

The following issues should be addressed in the revised manuscript. 

1) Other approaches to nonlinearity enhancement, e.g., the ones based on epsilon-near-zero materials should be briefly discussed. For instance, see Suresh, S., et al. ACS Photonics, 8(1), 125-129 (2020).   2) In Introduction (lines 42-43), the author mentions various modes. Which of them are especially suitable for the particular purposes of this work and why? Which of them are achievable in the designed metasurfaces at subwavelength scale?   3) In the last paragraph of Introduction, it is worth to add the phrase "In contrast with the previous studies, ..." or "In comparison with the earlier studies ..." [and then clearly state the differences as compared to the previous work].   4) It should be explained which relation has the proposed design to the field of nanomaterials, while it is linked rather with microwave frequency range.   5) The used resonators seem to be deeply subwavelength. Perhaps, larger P and larger other sizes might be more suitable to mitigate requirements to  fabrication, whereas the unit cell of metasurface may remain well subwavelength. Please, comment on it.   6) The paper suffers from the lack of experimental data. Therefore, the required experimental setup, required fabrication steps, and possible risks that may worsen performance as compared to the designed one should be briefly discussed.   7) The earlier works on the deeply subwavelength resonators, e.g., Alici, et al. Applied physics letters, 91(7), 071121 (2007) and Serebryannikov, et al. IEEE Transactions on Antennas and Propagation, 68(7), 5071-5081 (2020) should be compared in terms of the dominant physics and extent of miniaturization with the design proposed in the present manuscript.   8) In Section 3 (Results), it is worth introducing separate subsections that are dedicated to Frequency-Domain results and to Time-Domain results.   9) Explain why Frequency-Domain results are needed prior to the study of nonlinear effects by using Time-Domain simulations.   10) Check the correspondence of Fig. 2  and the text dedicated to this figure, e.g., lines' colors and style etc. Perhaps, it should be done for other figures as well.   11) Explain for wide reader what specific is in the results in Figs. 2 and 3 that may indicate their usefulness/importance for the analysis of nonlinear wave processes.   12) Lines 259-260: the authors mention a linear dependence. Whether it happens always or not? Does it have restrictions?   13) The term "spectra in the time domain" sounds not good, because "spectrum" is commonly associated with Frequency Domain. Please check it and try to use a more suitable terminology throughout the paper.  14) It should be clarified for wide reader what is input and what is output in the used Fourier transformation.   15) Explain to wide reader the meaning of the term "total conversion efficiency" that is used by the author.   16) I would recommend to repeat simulations for Fig. 7(b) in order to have more details (points) near w2=0.5 mm. 

Style, terminology, etc.

Line 101: it seems that "Ag" should be instead of "plasmonic material"

Line 232: "(" at the beginning of the formula should be omitted

Lines 332, 334: "g=" should be omitted before "0.025", "0.35", "0.4" and "1.1" 

to avoid repetitions

The literature list should be double checked for completeness of the provided information. For instance, journal name is not given in ref. 30.
